# Modelling sensory attenuation as Bayesian causal inference across two datasets

**Anna-Lena Eckert**[1]*, **Elena Fuehrer**[2], **Christina Schmitter**[3], **Benjamin Straube**[3], **Katja Fiehler**[2], **Dominik Endres**[1]

**1** Department of Psychology, Theoretical Cognitive Science Group, Philipps-Universität Marburg, Marburg, Germany, **2** Department of Psychology and Sport Science, Experimental Psychology Group, Justus-Liebig-Universität Gießen, Gießen, Germany, **3** Department of Psychiatry and Psychotherapy, Translational Neuroimaging Group, Philipps-Universität Marburg, Marburg, Germany

* eckertan@staff.uni-marburg.de

## Abstract

### Introduction

To interact with the environment, it is crucial to distinguish between sensory information that is externally generated and inputs that are self-generated. The sensory consequences of one's own movements tend to induce attenuated behavioral- and neural responses compared to externally generated inputs. We propose a computational model of sensory attenuation (SA) based on Bayesian Causal Inference, where SA occurs when an internal cause for sensory information is inferred.

### Methods

*Experiment 1* investigates sensory attenuation during a stroking movement. Tactile stimuli on the stroking finger were suppressed, especially when they were predictable. *Experiment 2* showed impaired delay detection between an arm movement and a video of the movement when participants were moving vs. when their arm was moved passively. We reconsider these results from the perspective of Bayesian Causal Inference (BCI). Using a hierarchical Markov Model (HMM) and variational message passing, we first qualitatively capture patterns of task behavior and sensory attenuation in simulations. Next, we identify participant-specific model parameters for both experiments using optimization.

### Results

A sequential BCI model is well equipped to capture empirical patterns of SA across both datasets. Using participant-specific optimized model parameters, we find a good agreement between data and model predictions, with the model capturing both tactile detections in *Experiment 1* and delay detections in *Experiment 2*.

### Discussion

BCI is an appropriate framework to model sensory attenuation in humans. Computational models of sensory attenuation may help to bridge the gap across different sensory

**Data Availability Statement:** All data underlying the described analyses are publicly available under https://osf.io/g5zbj/ (Experiment 1) and https://zenodo.org/records/2621302 (Experiment 2). All scripts for analyses and simulations are available

under https://gitlab.uni-marburg.de/fb04/ag-endres/tam-suppression.

**Funding:** This work was supported by the cluster project "The Adaptive Mind", funded by the Excellence Program of the Hessian Ministry for Science and the Arts. Open Access funding provided by the Open Access Publishing Fund of Philipps-Universität Marburg. The funders had no role in study design, data collection and analysis, decision to publish, or preparation of the manuscript.

**Competing interests:** The authors have declared that no competing interests exist.

modalities and experimental paradigms and may contribute towards an improved description and understanding of deficits in specific patient groups (e.g. schizophrenia).

## Introduction

Humans are constantly interacting with their environment, either processing information that arrives at sensory organs from the surroundings, or acting on the environment to bring about a desired outcome. It is crucial to distinguish between sensory events that are caused by changes in the environment from the consequences caused by one's own actions. Numerous studies indicate that self-generated and externally produced sensory information are processed differently [1–6]. In the auditory modality, tones that are produced via a button press elicit a dampened neural response compared to tones that are passively listened to [7, 8]. In the tactile domain, self-touch is perceived less intensely [9–12], and sensitivity to externally applied tactile stimuli on a moving limb is reduced [13, 14]. Sensory attenuation is ubiquitous in the animal kingdom [15–18]. Studies in *Drosophila* suggest presynaptic inhibition as the root cause of attenuated processing of self-generated sensory inputs during motor behavior [19].

Formal theories of sensory attenuation are rooted in engineering perspectives on motor control [2, 20], where the decision to move initiates a motor command. In optimal control theory, a forward model transforms the motor command into a prediction of the sensory consequences produced by that movement. Next, a prediction error between the predicted- and observed sensory consequences of the movement is computed. Whenever movement-induced sensations are in line with the predictions derived from the duplicate motor command (i.e., control theory prediction error is small), their processing is suppressed subsequently. If, for example, a motor command is sent to induce a saccade, a copy of the motor command is sent to visual areas, where compensatory mechanisms can counteract retinal displacement during the saccade [2]. *Sensory attenuation* of self-generated sensory events has been replicated across a variety of experimental paradigms and sensory modalities, at behavioral and neural levels [5, 9, 14, 21]. It is modulated by factors such as agency [6, 22], stimulus predictability ([4, 23, 24], but see [25]), and task- or feedback relevance [26–29] or feedback type (e.g. continuous or discrete, [30]). The experience of sensory attenuation seems furthermore important for feelings of agency and body ownership [22, 31, 32]. Supporting this, a study found increased sensory attenuation when attribution of agency is more difficult [33]. In schizophrenia, a disturbance of action-outcome monitoring may underlie body-related illusions [3, 32, 34, 35], as implied by observations of reduced sensory attenuation in patient samples and individuals at risk for psychosis [3, 36, 37].

The forward-model account of sensory attenuation suffers from some shortcomings [32]. For example, sensory attenuation is frequently observed in response to externally generated stimuli, for example, tactile stimuli presented by an experimenter to probe tactile perception. Since they are not self-generated, these stimuli cannot be interpreted as part of a forward model and should not be attenuated. Further, sensory attenuation has been shown to occur up to 400 ms before movement onset [38, 39], when it is implausible to be related to self-generated movement per se. Hence, the relationship between sensory attenuation and self-generated movement is likely not as straightforward as assumed by forward models and the phenomenon may be of broader importance than previously assumed [32].

theoretical developments extend optimal control theories and place many brain functions, such as perception and learning, under the mantle of Bayesian inference for optimal action

choices [40–42]. In *Active Inference*, Bayes-optimal behavior results from minimizing the prediction error between internal models and sensory information for perception, and the error between desired and obtained outcomes for action [43–45]. Real-world causes need to be *inferred* from noisy and multisensory information. The tight coupling between action and perception in active inference makes it a suitable candidate to study phenomena as sensory attenuation. Motor behavior in active inference is governed by "predictions, not commands" [43]: when a movement is initiated, the proprioceptive information of the stationary limb first needs to be down-regulated. This way, the internal belief can be realized [42]. Sensory attenuation may be the consequence of the down-regulation of proprioceptive precision prior to movement onset [32, 42], such that an initially false prediction about the current state of the body (e.g., predicting to move my arm while it is currently at rest) can be realized through peripheral reflexes despite conflicting sensory evidence (e.g. proprioceptive input from the resting arm). This account, in turn, has been criticized for its inability to account for intact perceptual monitoring during unexpected perturbations of movements [46, 47].

Similarly to Active Inference, *Bayesian Causal Inference* (BCI) is a normative framework of cognitive functions that centers around the idea of inferring the hidden causes of observations. Specifically, BCI assumes that the brain infers the cause of sensory inputs by selecting among competing causal structures [48]. Each causal structure is assigned a prior probability, which is updated using the congruency between the sensory inputs using Bayes rule. Besides offering a principled mechanism of causal inference, BCI can further be used to model the estimation of hidden variables [48]. BCI has been most extensively studied in the context of multisensory perception, where it can be thought of as a form of *competing priors* model (e.g.: inferring a common cause or two separate causes for multisensory inputs; [49–52]). Importantly, the principles of BCI appear to govern not only the perception of the outside world, but also feelings of body ownership [53–55] and motor learning [56, 57]. For instance, Samad and colleagues (2015) demonstrate that a BCI model can capture feelings of body ownership in the rubber hand illusion. Here, synchronous stroking of the rubber hand and the participant's hand increases the illusion of body ownership, while asynchronous stroking decreases the strength of the illusion. These classical findings were reproduced by a BCI model, which considered both temporal and spatial information from the proprioceptive, visual and tactile sensory modalities [55].

We here draw on the theoretical frameworks of Active Inference and BCI to develop a computational model of sensory attenuation. It is crucial for the brain to capture the causal structure underlying sensory inputs. Any changes in sensory information brought about by an individual's voluntary movement (i.e., internally-generated sensory information) tend to be highly predictable and will be processed in an attenuated manner. In contrast, sensory information caused by an important change in the (external) environment is potentially relevant and requires more processing resources. In our model, sensory attenuation therefore results from inferring an *internal* cause for sensory information. Whenever sensory information is labelled as *internally*, or self-generated, it is usually in line with the individual's movement-derived predictions about upcoming sensory information and processed in an attenuated manner in subsequent processing steps. Meanwhile, sensations where an *external* cause is inferred are represented with increased precision due to its potential relevance. We construct causal graphical models of the tasks and perform BCI by belief propagation, inspired by Active Inference. We apply the model to empirical data to uncover computational principles underlying sensory attenuation across two experimental studies [6, 23]. We first report qualitative simulations with psychologically plausible parameter settings. The model's parameters are subsequently recovered from fitting the model to the experimental data for each participant

individually and subsequently evaluated against a benchmark model with uninformed parameters.

To preview our results, BCI can capture psychophysical signatures of sensory attenuation qualitatively across two experiments, yielding that tactile stimuli predicted by the sensorimotor system [23] and visual-tactile stimuli proximal to one's own actions [6] are labelled as *internally* generated and hence attenuated. When reproducing task behavior with empirically derived model parameters, we obtain a good fit between psychometric properties of participant behavior (PSE) and model predictions.

## Methods

### Data

We fit our model onto data from two experiments [6, 23]. In the following, we will give an overview of the experimental procedures as relevant to the present work. All procedures are described in detail in the original publications [6, 23].

**Experiment 1.**   N = 32 healthy, right-handed human participants (N = 23 female, age range 19–30, mean: 22.56 ± 3.03 years) completed an experiment on movement-related tactile suppression at the University of Gießen, Germany. The customized experimental device consisted of a force sensor, a vibrotactile stimulation device attached to the proximal phalanx of the participants' right index finger, and two 3D-printed textured objects. Half of the objects' surfaces was smooth, and the other half had a texture with an even, square-wave pattern. The object's surface texture was determined by a large (5.08 mm, 40 Hz when moving across consistently at 203 mm/s) or small (0.85 mm, 240 Hz) spatial period. The participants' task was to stroke across the textured objects with their index finger at a constant velocity of 203 mm/s and report after the movement whether they had detected a vibrotactile stimulus prior to contacting the textured part of the object. Several practice trials with visual feedback were implemented to ensure participants adhere to the target motion speed. The empirical motion speed was 203 mm/s ± SD 35 mm/s (range: 110–329 mm/s, see original publication, supplementary Fig 6). If participants deviated considerably from the prescribed movement, the trial was repeated, and verbal feedback was given. The brief probe stimuli (100 ms), delivered via the vibrotactile stimulation device, were presented at frequencies of either 40 Hz or 240 Hz. The participants' hands were not directly visible to them, but the scene and the hand position were presented on a computer screen and viewed via a mirror. Five pseudo-randomized blocks were presented in the experiment, with one block per movement condition 2 (probe frequencies) by 2 (object frequencies), and one baseline condition. For further information, see Fig 1A and the original publication [23].

**Experiment 2.**   N = 23 healthy, right-handed participants (N = 12 female, age range 20–35, mean age 26.43 ± 3.99 years) completed the experiment at the University of Marburg, Germany. Participants were instructed to hold the handle of a custom movement device for each trial. The device's handle could be moved from a neutral starting position to the right and back. In *active* trials, the participant actively moved the device's handle from neutral to right and back. In *passive* trials, the participant's hand was moved by the device via air pressure. The device and the participant's hands were covered from their view, but recorded with a high-speed camera and played back onto a computer screen. Participants saw video recordings of their own hand performing the movement (*self* trials) or a previously recorded video of another person's hand performing the movement (*other* trials) that were directly coupled to their own movement. The experimental manipulation consisted of randomly inserted, variable delays (delays = [0, 83, 167, 250, 333, 417ms]). A 2 (agency: active vs. passive) by 2 (hand identity: self vs. other) by 6 (delay level) design resulted. All participants completed three sessions

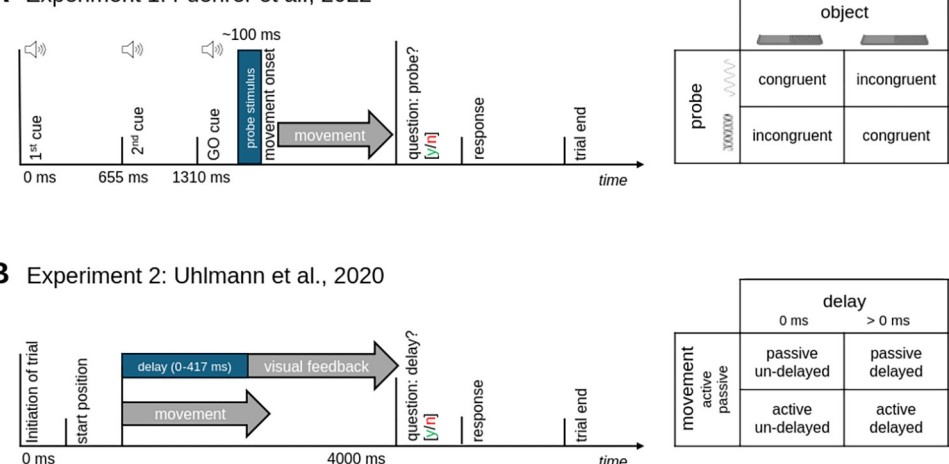

**Fig 1. Experimental procedures. A.** *Top*: Trial schematic *Experiment 1*, adapted from Fuehrer et al., 2022. Participants were seated in front of a desk with a textured object, force feedback device and sensors. A vibrotactile stimulation device was attached to their right index finger. Participants made a stroking movement across the textured object (40 Hz or 240 Hz at constant velocity of 203 mm/s) and were asked to detect a vibrotactile probe delivered via the stimulation device before contacting the textured part of the object. *Right*: Probe and object frequency were designed to be either congruent or incongruent. For details, please refer to the original publication. **B.** *Top*: Trial schematic, *Experiment 2;* adapted from Uhlmann et al., 2020. Participants held the lever of a movement device. Participants performed arm movements (*active* trials) or their hand was moved by the device (*passive* trials). Participants had to detect experimental delays (0, 83, 167, 250, 333, 417 ms) between their movement and a video of their hand. Note that we omitted the experimental manipulation of hand identity for simplicity (see Methods, Experiment 2). For more details, please refer to the original publication.

of the experiment (one during a preparatory session, two fMRI sessions). We here consider the behavioral data of all sessions. We do not consider the experimental manipulation of hand identity here since it did not have a significant effect on behavior, with an insignificant main effect of hand identity on psychometric function threshold and slopes $F(1,22) = 0.185$, $p = 0.671$ and no interaction between hand identity and agency (active vs. passive), $F(1,22) = 0.744$, $p = 0.332$ (cmp. [6]). Each run contained 48 trials with an active and passive block, respectively, and randomized self vs. other visual feedback. In each trial, participants were instructed to perform the hand movement actively or that their hand was moved passively, and they had to indicate whether there was a delay between their own movement and the video feedback of their movement (see Fig 1B).

## Model

A graphical Bayesian Causal Inference model was developed to capture empirical patterns of sensory attenuation across the two experiments. A graphical model was chosen because of its usefulness for characterizing probability distributions, which we use as a proxy for the inferential mechanisms underlying decision-making in an experimental trial. A sequential model architecture furthermore aligns closely with the generative models commonly used in the context of Active Inference. From Active Inference, we also draw on belief propagation algorithms and free-energy based parameter updating methods [40, 42, 58, 59]. We use a custom Python suite for sum-product belief updating with exponential family distributions [58, 60]. For further information on the belief propagation algorithm, please see the S1 File.

The model for experiment 1 is a Hidden Markov Model (HMM), with a series of *unobservable* variable nodes forming a hidden Markov chain, and one or more *observable* nodes attached to each hidden node (Fig 2A). A set of HMMs with a causal inference variable

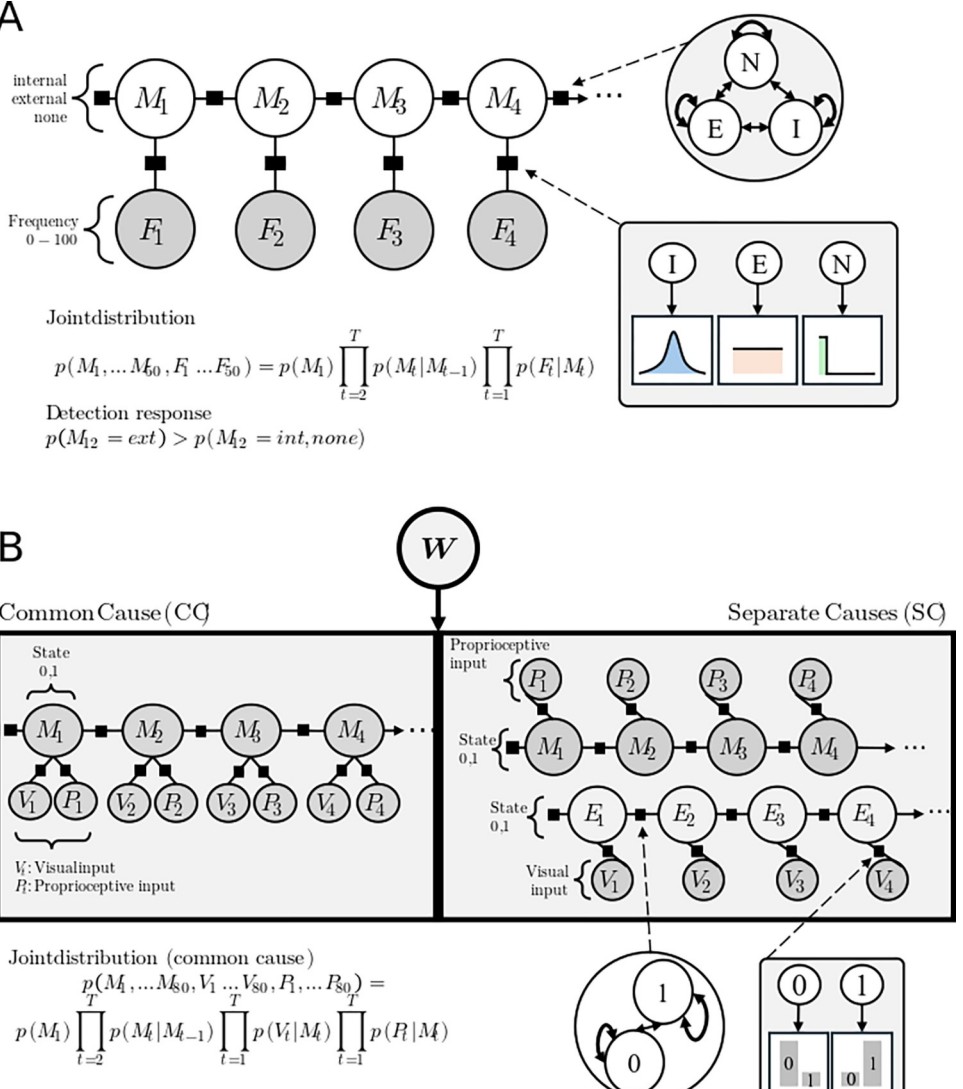

**Fig 2. Graphical models. A.** Graphical model for *Experiment 1*. Hidden nodes form a Markov chain that can take on values {*internal*, *external*, *none*}. Observable nodes $F_t$ (filled circles) can process frequency inputs between 0–100 Hz. The joint probability distribution over the network is given by the prior and the product of all transition- and emission nodes. A detection response ($r_t = 1$) occurs when the normalized posterior probability of the *external* cause exceeds the normalized posterior probability of an *internal* or *none* cause. **B.** Graphical model for *Experiment 2*. Under a common cause model (left box), proprioceptive $P_t$ and visual information $V_t$ are caused by the same hidden factor $M_t$, whereas in a separate cause model (right box), different causes $M_t$ and $V_t$ are inferred for the proprioceptive $P_t$ vs. visual $V_t$ inputs. The switching node **W** compares the marginal log likelihood of the observations under a common- vs. separate cause assumption and determines the dominant model in each trial. The joint probability distribution of the network is given by the priors and the product of all transition- and emission factors. A delay is detected when the posterior marginal probability of the common cause model (CC) given the observations exceeds the marginal posterior of a separate cause (SC) model.

switching between a common underlying cause and separate causes is developed for experiment 2 (Fig 2B).

**Experiment 1.** The model consists of a Markov chain of 50 hidden variable nodes. A Markov chain of length $T = 50$ was chosen to match the experimental trial duration, with one time-step representing 10ms in the experiment. These nodes can take on three possible values

(variable range: cause $C \in \{internal, external, none\}$). One observable node is attached to each hidden node. Observable nodes can take on values between 0 and 100 (observation $O \in \{0-100\}$), representing the observable frequencies elicited by the experimental stimuli (Fig 2A). The connections between hidden and observable nodes are determined by likelihood factor nodes $L$. The likelihood mediates the relationship between an observed frequency and the likely cause via probability distribution $P(O \lor C)$. Three assumptions were made to build the likelihood distribution for factor nodes $L$ (see Fig 4A): 1) If no frequency is observed, observations around a 0 frequency are most likely. 2) If the cause is *external*, all frequencies are equally expected; 3) If the cause is *internal*, the prediction is normally distributed around a frequency determined by movement speed. The variance of the likelihood function σ can be interpreted as the combination of motor- and sensory noise. Furthermore, transitions between hidden nodes are governed by a probability distribution $T$. Parameter values for the transition nodes were task-inspired, with e.g. the *internal* inference being more stable than the *external* or *none* state.

Sensory attenuation occurs when an observed probe frequency is inferred as *internally caused*, which is likely when the frequency is in the range of the expected frequencies given the current movement policy and object texture. In other words, in our conceptualization, sensory attenuation means incorrectly attributing an externally caused sensory event (here: the vibro-tactile stimulation) to an internal cause (here: the planned finger movement across the grating). Once inferred as internally caused, the observed frequency is processed in an attenuated manner due to the predicted sensory consequences of the movement. It follows that in the experiment, a probe can only be detected if it is correctly inferred as externally generated. Our model hence delivers the probability estimate $P(M_t = external)$ that should approximate detection events $r_t = 1$.

For qualitative simulations, we directly converted the sequence of frequencies within one trial into observations that were then input into the model's observable layer (Fig 4B). For example, in a high-frequency congruent trial, the observed frequencies are converted into a list like (shortened) [0,0,0,30,0,0,0,30,30,30], where the first instance of a frequency of 30 (surrounded by zeros) represents the probe presentation, and a longer phase of observing the frequency represents object contact (padded to achieve a time series of length $T = 50$). In line with the experiment's two by two design, frequency series represented congruent (low frequency probe and low frequency object, high frequency probe and high frequency object) or incongruent (low frequency probe and high frequency objector high frequency probe and low frequency object) probe-object frequency pairings.

The model is furthermore fit onto the empirical data. Python, and the SciPy package for optimization (v. 1.10.1., Virtanen et al., 2020) are used to identify optimal model parameters (likelihood $L$ and transition $T$ distributions) for each participant. The cost function is given by

$$c = q \cdot \log\left(\frac{q}{p(ext)}\right) + (1-q) \cdot \log\left(\frac{1-q}{1-p(ext)}\right) \tag{1}$$

With $q$ equal to the average of all detection events, $q = \frac{\sum r_t = 1}{N}$ where $N$ is the number of trials. Minimizing $c$ will yield model parameters that lead to an optimal match between probe detection responses ($r_t = 1$) and the probability of the external cause $P(C = external)$ for each trial type in the movement condition. In addition to the divergence between the model and the training data, a cost term was added for two endpoint constraints. At the beginning of the time series, a hidden None state (i.e., no stimulation) was enforced to be consistent with the training data ($p(M_5 = None) \sim 1$). Towards the end of the timeseries, the cost function enforced an *internal* state ($p(M_{40} = internal) \sim 1$), to account for participants having made contact with the object by this point, ensuring certainty about an internal state. All conditional probabilities were

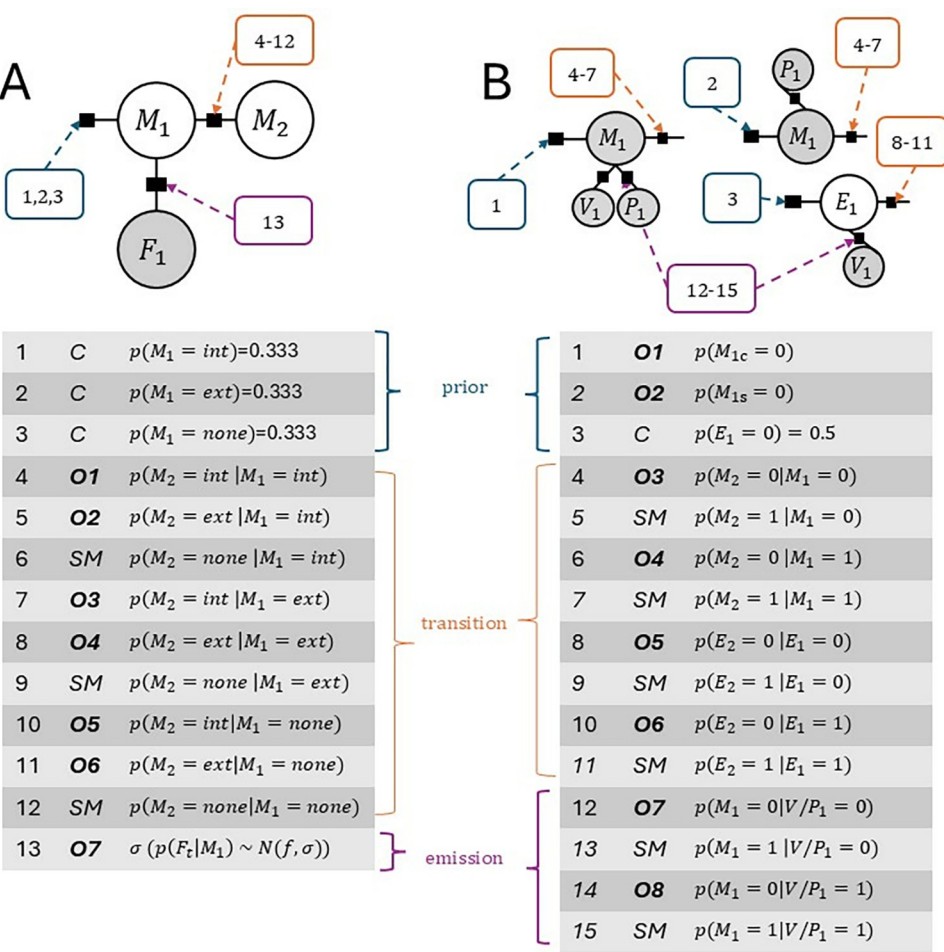

**Fig 3. Parameter overview. A.** Parameters of model for Experiment 1. C: constant, **Oi**: Optimized, free parameter number i, SM: obtained via exploiting normalization condition of conditional probability distributions. There is a total of 13 parameters in the model (see top factor graph). Initial recovery indicated that the priors on the first hidden state ($p(M_1)$) do not influence detection behavior sufficiently. They were held constant at uninformative levels ($p(M_1 = [int, ext, none]) = 0.33$). As a result, there are 7 free parameters for modelling Experiment 1 (see tabular overview). **B.** Parameters for model of Experiment 2. The graphical network contains a total of 15 parameters, of which 8 are free parameters. As before, remaining parameters are computed via conditional distributions (SM).

represented by their logits for the purpose of optimization. Their range was restricted to $(-3,3)$ and the starting values were chosen within this range, too. The optimizer used the Powell method [61] and tolerance was set to $1e^{-12}$. For each participant, the optimization was initiated 10 times, where the optimizer's starting values were jittered within the predefined bounds for each optimization to ensure the results' robustness. The best set of parameters (i.e., the parameters minimizing cost) was chosen for each participant. There were 7 free parameters in the graphical model. For an overview of the parameters to be optimized for the models see Fig 3A. A parameter recovery served to validate the model (Supplement S2 in S1 File).

## Model comparison

We compared the full, causal model to two alternative models, a bias-only model and an uninformative model. In the bias-only model, participants are expected to show idiosyncratic response patterns in the probe detection task, which are not directly related to the

experimental manipulation. The uninformative model predicts equal detection probabilities for all three causes (p = 0.33). To compare the models M, we approximate the model evidence, or the log probability of the data D under the model and a tuple of its parameters Θ:

$$\log p(D|M) = \log \int d\Theta p(D|\Theta, M) p(\Theta|M) \qquad [2]$$

After evaluating the model evidence for all models, we compare model evidence differences. As usual in Bayesian approaches, the integral over model parameters is difficult to evaluate. We hence approximate the model evidence using the Laplace approximation LAP [58, 60]. The LAP is given by [60].

$$p(D|M) \approx \underbrace{\log(p(D|\Theta^*, M))}_{\log-\text{likelihood}} + \underbrace{\log(p(\Theta^*|M))}_{\log-\text{prior}} + \underbrace{\frac{\dim(\Theta)}{2}\log(2\pi) - \frac{1}{2}\log(|\mathbf{H}|)}_{\log-\text{posterior volume}} \qquad [3]$$

We evaluate the Hessian $\mathbf{H}$ with autograd [[62], v. 1.0] and find the optimal model parameters $\Theta^*$ using the L-BFGS-B optimizer from the scipy.optimize library [[63], v. 1.14.0]. The LAP is a more accurate approximation to the model evidence than the BIC. Endres, Chiovetto and Giese (2013) show that the BIC performs comparably to the LAP only in the limit of datasets larger than ours. Therefore, LAP is a more exact approximation to the Bayesian model evidence than the BIC in the present case. As a global measure of model fit, we furthermore report the Kullback-Leibler divergence given by

$$D_{KL}(p_{data}\|p_{model}) = p \, log\left(\frac{p}{p_{est} + \epsilon}\right) + (1-p) log\left(\frac{1-p}{1-p_{est} + \epsilon}\right) \qquad [4]$$

Where $p$ is the model-derived probability of detecting a probe, $p_{est} = \frac{\sum r_t == 1}{T}$ and $\epsilon = 1e^{-100}$ is a constant to avoid numerical underflow.

**Experiment 2.** In the experiment, participants were asked to detect delays between their own movement and a video recording of the same movement. We framed this problem in terms of causal inference, where participants can infer either a common cause or a separate cause for the visual and proprioceptive information elicited by the movement and experimental stimuli. A delay detection corresponds to inferring a higher posterior probability for the *separate cause model*, whereas smaller potential delays may be suppressed as a function of inferring a common cause for visual and proprioceptive information. The common cause model consists of a Markov chain of 80 hidden variable nodes. A Markov chain of length $T = 80$ was chosen to represent the six levels of delay manipulation (transformed in deci-seconds (ds)). Hidden nodes can take on the values $H \in \{0,1\}$, representing the neutral- and right lever position, respectively. Attached to each hidden node in the common cause model are two observable nodes representing visual and proprioceptive sensory inputs. Both node types can take on values 0 or 1, $V \in \{0,1\}$; $P \in \{0,1\}$ (Fig 2B). Experimental delays are formalized by observing series of sensory input that are shifted against each other by the amount of timesteps corresponding to the experimental delay. Sensory information and hidden state share a strong correspondence in the common cause model due to the likelihood distribution. Whenever the hidden state is in the "1" position, it is highly likely that the proprioceptive and visual inputs are in the "1" position as well. If this is not the case, the common cause model becomes more unlikely a-posteriori. The separate cause model consists of two separate Hidden Markov Chains with 80 timesteps, where only one observable node is connected to any hidden node. The first chain represents the visual observations and their causes, while the second chain

models the proprioceptive inputs and their causes. In contrast to the common cause model, proprioceptive and visual information can be caused by different factors in the separate cause model. In other words, underlying the separate causes model is the assumption that the proprioceptive information was caused by an active or passive hand movement, while the visual information was caused by the experimenter playing a video clip of the movement instead of the real-time video feedback of the movement. In each trial, a model comparison is performed, where the posterior probabilities of the common cause model and the separate cause model are weighted against one another. Whenever $P(M = separate \lor D) > P(M = common \lor D)$, a delay is detected.

To qualitatively simulate the patterns of sensory attenuation in this experiment, visual and proprioceptive sensory inputs were formalized as identical streams of model observations, or as two streams of information shifted against one another with varying levels of delay (Fig 4C). Observations were fed to the common cause- and the separate cause model, respectively. We then compared the posterior probability of the data, given the models, with higher posterior probabilities determining the winning model and hence perception.

SciPy-optimize [63] was used to fit the model on the experimental data and identify optimal model parameters (i.e., likelihood parameter, mediating the relationship between hidden and observable nodes; and transitions between hidden nodes) given the data. A total of 15 parameters are considered in the optimization (see Fig 3B), consisting of priors, transitions and emissions for both the common and separate cause models, respectively. The emission model is shared between the common- and separate causes chain. Similar to *Experiment 1*, the cost function is given by:

$$c = \mathrm{p}(sep) \cdot log\left(\frac{\mathrm{p}(sep)}{q}\right) + (1. - \mathrm{p}(sep)) \cdot log\left(\frac{1. - \mathrm{p}(sep)}{1. - q}\right) \qquad [5]$$

With $P(sep)$, the posterior probability of the separate-cause model, and the average of all detection responses $q = \frac{\sum_{r_t=1}}{N}$ where $N$ is the number of trials. Minimizing $c$ will yield model parameters that lead to an optimal match between delay detection responses ($r_t = 1$) and the posterior probability of the separate cause model, $P(sep)$. To ensure that all probabilities are between 0 and 1, we optimize the logit of the parameters. Starting values for the logit was set to be between (-1, 1). The optimizer method used was Powell with a tolerance of $1e^{-8}$. For each participant dataset, the optimization was initiated 10 times, and the best resulting set of parameters was selected (i.e., the parameters that minimize cost across the 10 results).

A crucial experimental manipulation is the distinction between trials with an active vs. a passive arm movement. To explore the effect of this experimental manipulation, we fit the model separately to active vs. passive trial data to simulate model-derived psychometric functions. This was done to examine whether our model can reproduce the main empirical finding of Uhlmann et al., 2020 of heightened delay detection thresholds in active trials. As before, a parameter recovery was performed to validate the model. Details on the procedure can be found in supplement S2 of S1 File.

## Model comparison

We compare the full causal model to two competing models commonly used in multisensory integration research: forced fusion and forced separation [50, 57]. In a forced fusion model, the proprioceptive and visual information will always be attributed to the same hidden cause. Contrarily, in forced separation, the visual- and proprioceptive information streams are always attributed to separate underlying causes. We furthermore introduce a bias-only model as

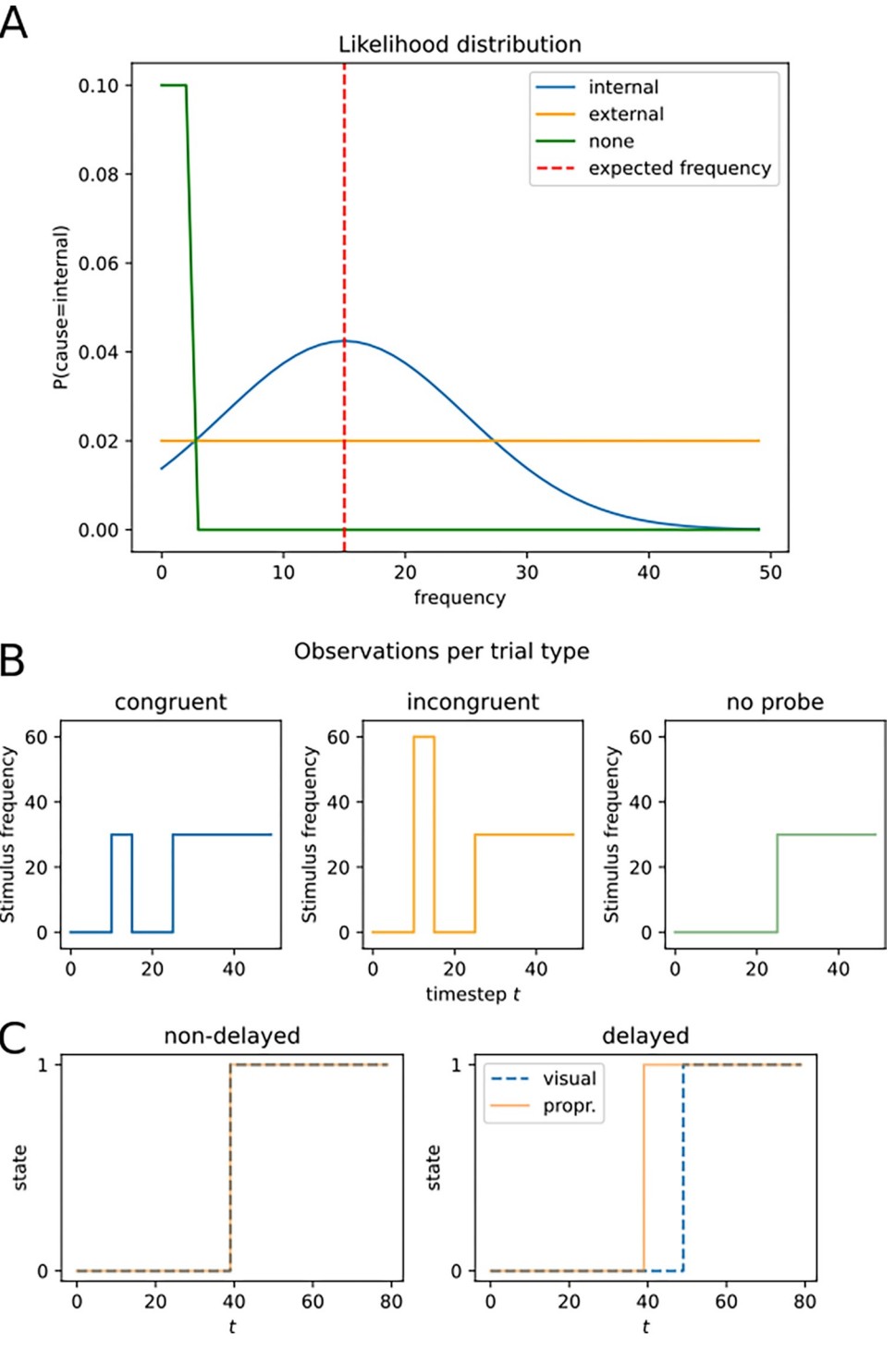

**Fig 4. Model specifications. A.** Likelihood function for *Experiment 1*. If the *none* cause is inferred, frequencies around 0 Hz are expected. Under the *external* cause, all frequencies are equally unpredicted. If the *internal* cause is inferred, the mean predicted frequency is normalized around the frequency expected given a movement policy. **B.** Experiment 1, observation schemes for each trial type; probe congruent to object (left), probe incongruent to object (center), no probe (right). **C.** Experiment 2, observation schemes per trial type, non-delay trials (left), where visual and proprioceptive inputs follow the same temporal pattern (i.e., shifting in state at the same time), vs. delayed trials, in which the visual information is shifted against the proprioceptive input in time.

before, where participant choices are a function of their idiosyncratic response biases. The Bayesian model evidence is approximated using the LAP, as described previously [58, 64].

## Results

### Experiment 1

Simulations. We simulated a time series of 50 nodes, with internal cause (internal, external, hidden) as a hidden layer, and frequency observations (0–100) at the observable layer. A series of frequency observations, contingent with the four experimental conditions, was developed for the model. Qualitative results yield a good match between the probabilities predicted by the model and participant behavior. Congruent probes, i.e. those matching the predicted sensory consequence of the movement, are inferred as *internally* caused (Fig 5A). We assume that sensory attenuation of highly predictable (i.e., congruent) probes results from inferring an *internal* cause for the probes. In contrast, incongruent probes are inferred as *externally* generated by our model (Fig 5B). This pattern is mediated by the nature of the probability distribution contained in the emission nodes, where a given frequency that is expected under the current motion policy determines the mean of a normal distribution. When the probe's frequency matches this mean expectation, an internal cause for it is inferred. In the incongruent case, the mean expected frequency is different from the one experienced during probe presentation, and hence an external cause is inferred for it. Under an external cause, all frequencies are equally (un)likely, which is why the model prefers it for unexpected frequencies. Hence, the model qualitatively captures the effects of stimulus predictability on underlying causal inference. We next fit this model to the experimental data by Fuehrer et al., 2022.

**Data fitting.** Optimization was performed for each participant individually. The optimization converged for all individual datasets. The objective function was evaluated on average 158 times (range: 112–210, standard deviation: 22.5). The optimizer converged after on average $N_{fev}$ = 333 (min: 294, max: 434, standard deviation: 25.25) evaluations of the objective function. A total of 12 parameters was optimized per participant. Optimal parameters on average consisted of uninformative priors over causes ($P$(*cause* = (internal,external,none) = 0.33)), and a transition matrix with a stable internal–internal transition and an unlikely transition from the *none* cause to an *internal* or *external* cause ($P$(*internal*|*internal*) = 0.68; $P$(internal|none) = 0.26; $P$(external|none) = 0.17, see Fig 6B). The average noise parameter (sum of motor and sensory noise) was $\sigma$ = 20.61 (Fig 6C). We next compared empirical patterns of probe detections against detection events as predicted by our model ($P$(*external*)>$P$(*internal*),$P$(*none*)). The model is capable of capturing detection events particularly in the incongruent conditions (Fig 6A). It predicts significantly less detection events in the congruent conditions compared to the empirical data. We next initialize the model for each participant with their individual optimized parameters. Simulating each trial type, the model yields a clearer distinction between causes at the time of probe presentation in incongruent trials (Fig 7C and 7D for an example) compared to congruent trials (Fig 7A and 7B).

Model comparisons yielded an advantage for the full causal model. The evidence difference per participant ($\Delta$ LAP causal–LAP bias) was 1.20 (± standard error 0.15), indicating an advantage for the full causal model over a bias-only model on the participant level. This translates into a probability ratio of 4.9x10$^{16}$ on the group level. This indicates strong superiority of the causal model on the group level. The bias-only model held an advantage over the uniform model ($\Delta$ LAP bias—LAP uniform), with a group-level average of 28.54 ± standard error 5.46 (Fig 8). As a measure of global fit of the full model, we computed the KL-divergence between model-derived and empirical response distributions. Mean KL divergence for Experiment 1 was $KL_{mean}$ = 1.262 nats (Fig 14A). One outlier (KL = 12.56 nats) was related to a higher-than average detection frequency, and a lack of variance in the incongruent condition (see S2 in S1 File) Finally, prediction

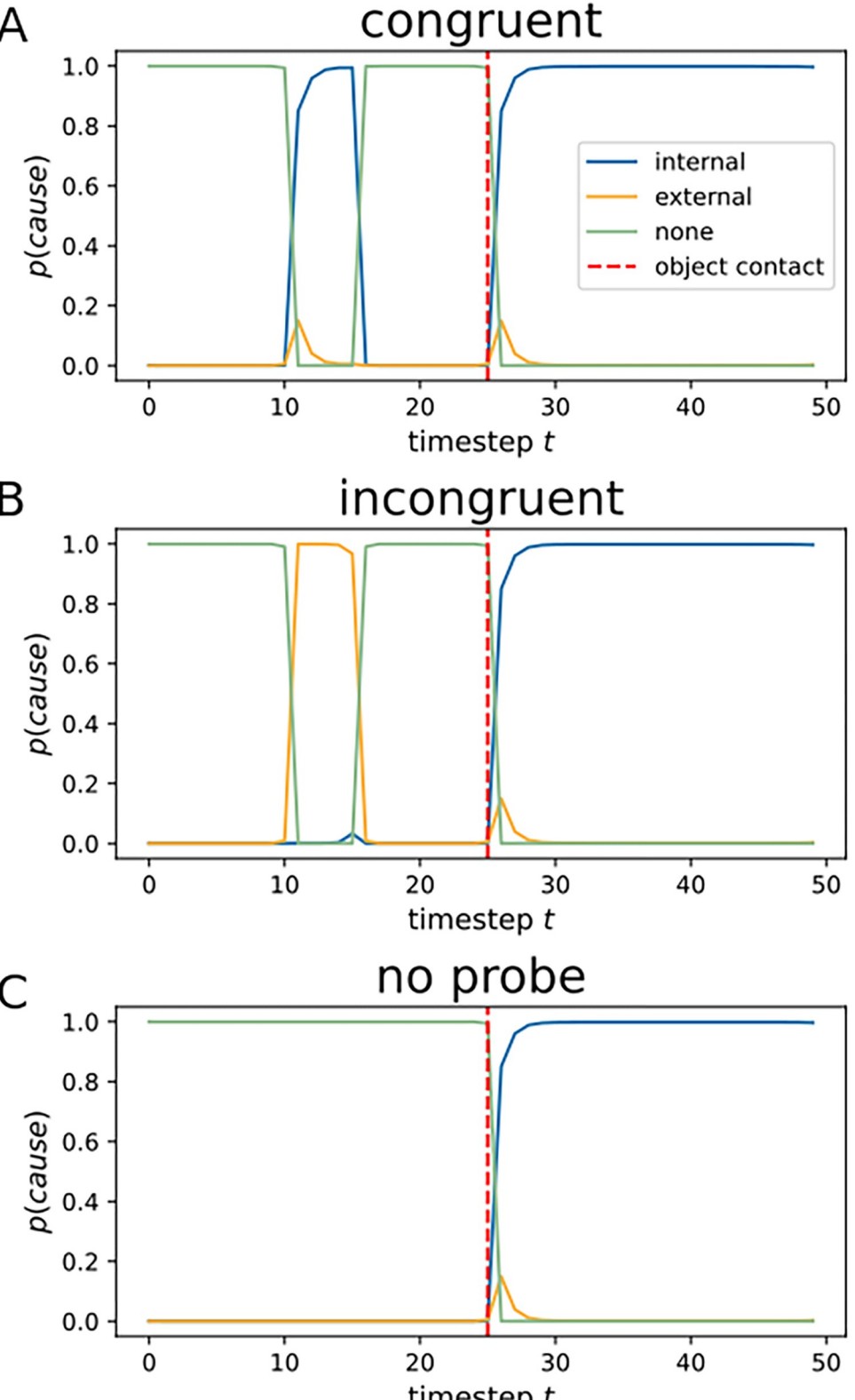

**Fig 5. Qualitative simulations (Experiment 1). A.** Simulation of a congruent trial, where an internal cause is inferred for the probe. **B.** In the incongruent probe trial, an external cause is inferred for the probe. **C.** In the trial without a probe, no external or internal cause is inferred.

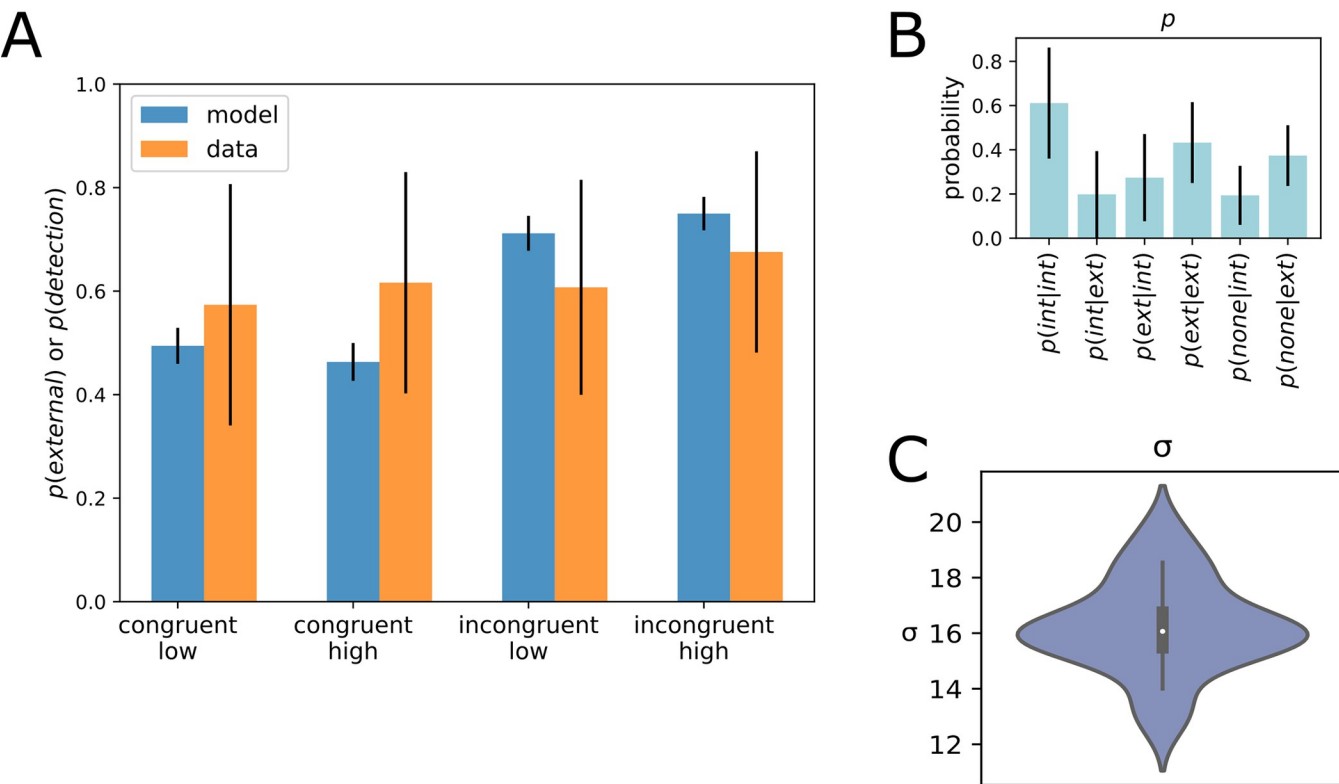

**Fig 6. Results of optimization, Experiment 1. A.** Detection probabilities per trial type in the empirical data (orange) vs. detection probabilities per trial type as predicted by the model (blue). Congruent low: congruent, object and probe have a low frequency. Congruent high: Congruent, both object and probe have a high frequency. Incongruent—low: Incongruent, probe has a low frequency, object has a high frequency. Incongruent—high: incongruent, probe has a high frequency, object has a low frequency. Error bars represent standard deviations of model predictions (blue bars) and empirical data (orange bars). **B.** Average optimized transition parameters yield biased transitions, with a sticky internal cause. Considerable variance between subject is present especially for the transition parameters *p(int|int) and p(ext|int)*, as indicated by the error bars (black) representing standard deviations of the parameter estimation. **C.** Violin plot of average optimized *σ* (motor noise) parameter.

recovery showed that the fitting procedure does not introduce systematic biases and captures the relationship between data and model-based detection probabilities. Interestingly, the trial type drives certain parameter's recoverability (see supplementary chapter S2, S15 Fig in S1 File).

## Experiment 2

**Simulations.** We simulated a time series of length $T$ = 80, with hidden causes (position 0 or 1) forming a chain of hidden nodes, and observable nodes for visual and proprioceptive sensory inputs. Consistent with the experimental delays, a series of sensory observations for the model's observable nodes were developed. In a first qualitative simulation, the model yielded results in line with the empirical results, where for smaller delays, a common cause model is inferred, and a separate cause model is inferred for larger delays (Fig 9A). Similarly, the model was used to simulate the task-specific psychometric function (Fig 9B), showing increasing detection responses with higher delays.

**Data fitting.** The optimization procedure yielded participant-specific values for twenty parameters, with 10 parameters each for the common- and separate-cause models, respectively. Averaged parameter values are shown in Fig 10. Parameter values indicate slight differences in average optimal parameters between the common- and separate cause models, with e.g. an increased prior on starting state 1 in the separate cause model (separate cause model: *P*

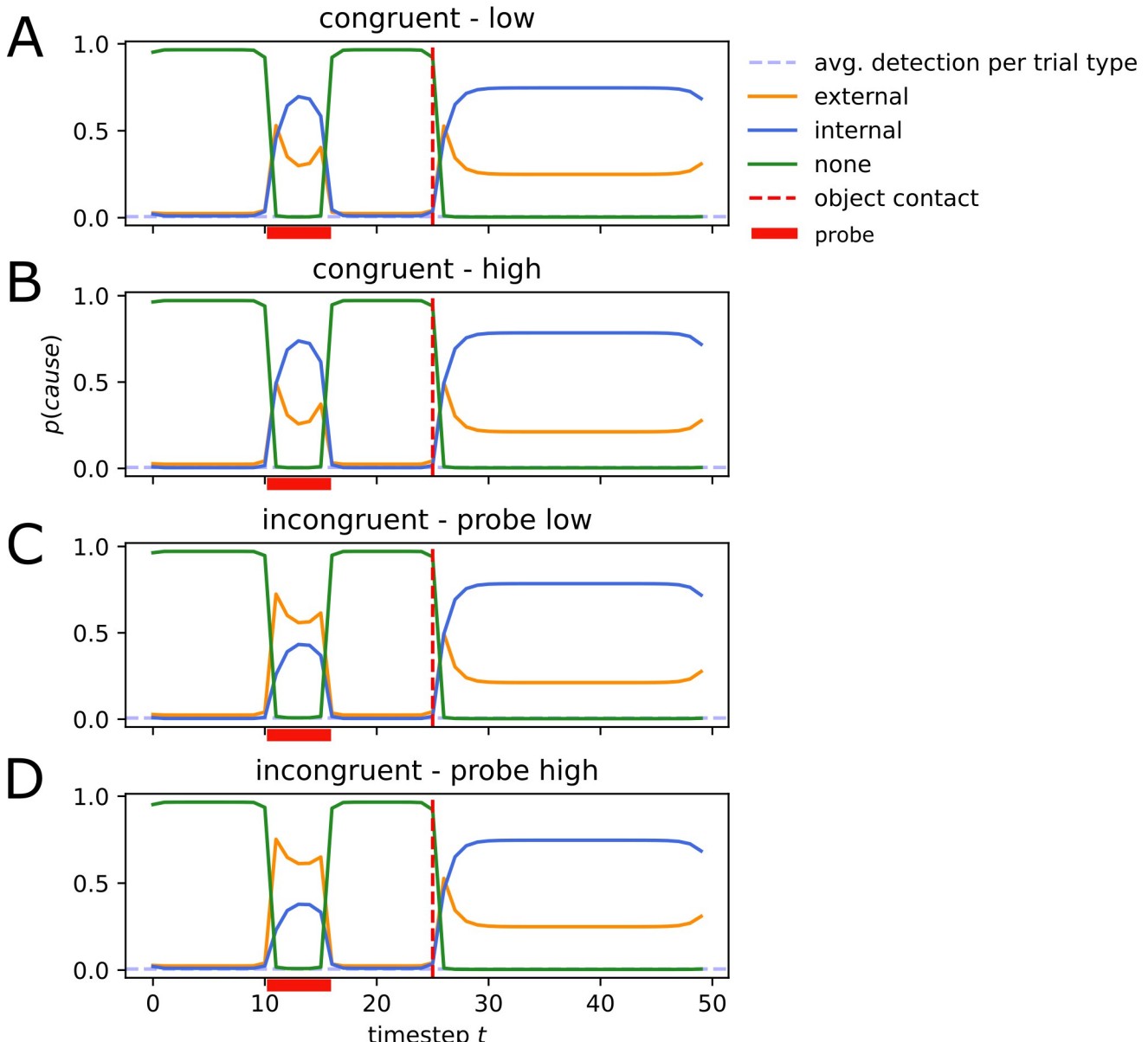

**Fig 7. Average simulated time series with optimized parameters (Experiment 1).** Simulation of all trial types (congruent low (**A**), congruent high (**B**), incongruent–probe low (**C**) and incongruent–probe high (**D**)) with average optimized parameters. The probe is presented between timesteps 10 and 15 in the timeseries (red bar on x-axis). In the congruent trials, the probe is mistaken as internally generated (*p(internal) > p(external))*. In incongruent trials, the probe is labelled as externally generated (p(external) > p(internal)), albeit with some uncertainty.

(1) = 0.54, common cause model: $P(1) = 0.50$) and an increased transition probability between states 1 and 1 over time in the separate cause model (separate cause model: $P(1-1) = 0.64$; common cause model: $P(1-1) = 0.52$). We next compared empirical delay detection event frequency per delay with the model-derived detection events, as approximated by the probability of inferring a separate cause for sensory information. Inputting the optimized parameters per participant into the graphical model, we computed the probability of inferring separate causes

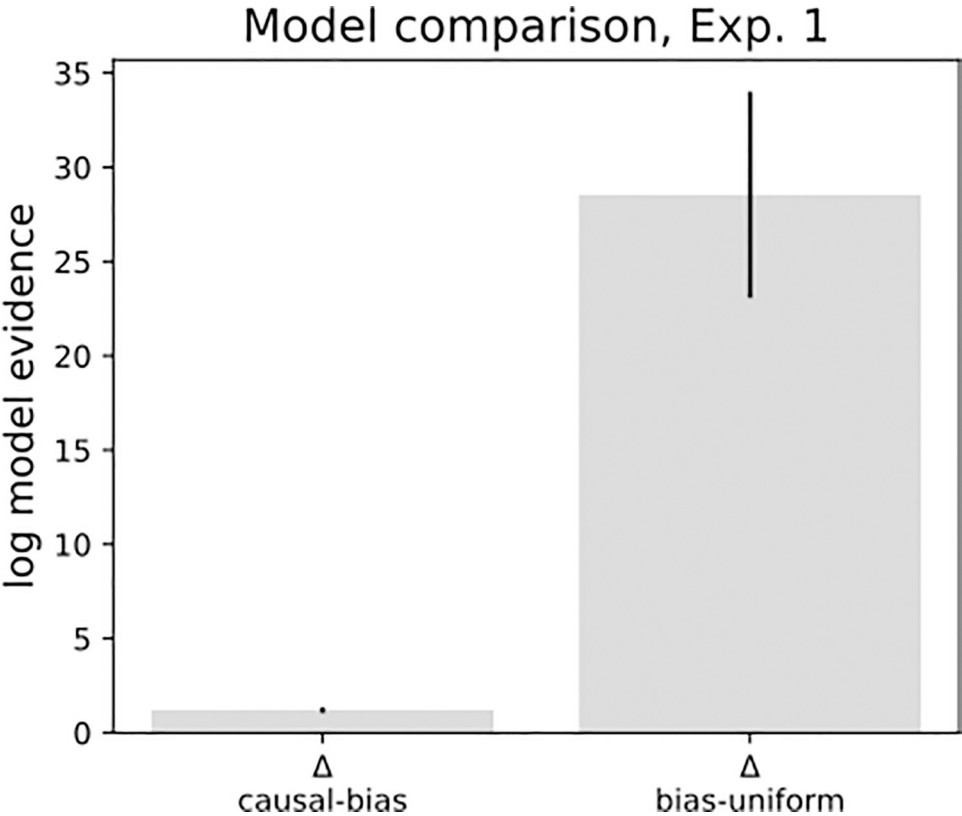

**Fig 8. Model comparison, Experiment 1.** The full causal model is compared to a model where responses are predicted exclusively by individual biases (*bias*) and one with uninformative parameters (*uniform*). Displayed are probability rations, or differences (Δ) in LAP. Positive values indicate an advantage for the causal model compared to the bias-only model, and an advantage over the bias model over a uniform model. Black lines represent standard errors.

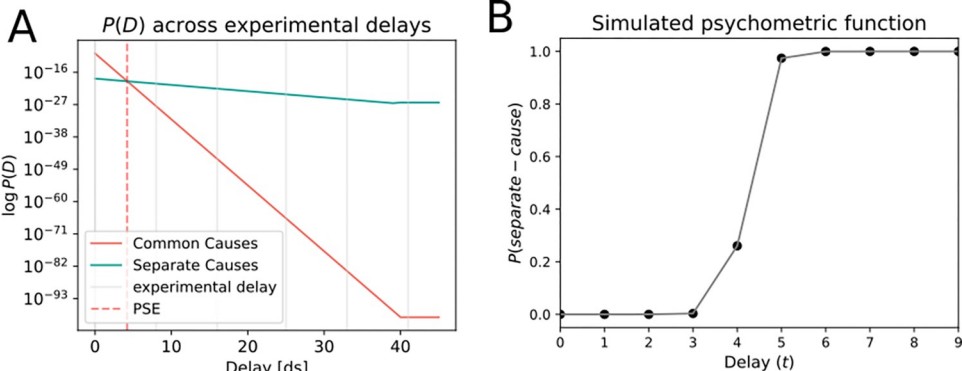

**Fig 9. Simulated PSE and psychometric function (Experiment 2). A.** The simulated point of subjective equality (PSE)is represented by the red dashed line, indicating the PSE is located at a delay of 5 ds. In smaller delays, the common cause model (coral line) has a higher marginal probability. For larger delays, the separate cause model (turquoise line) is inferred as more likely. **B.** Simulated psychometric function for Experiment 2. The probability of inferring separate causes for the visual and proprioceptive information increases with increasing delay duration (Unit of time: deci-second, ds).

## Average optimized parameters, Exp. 2

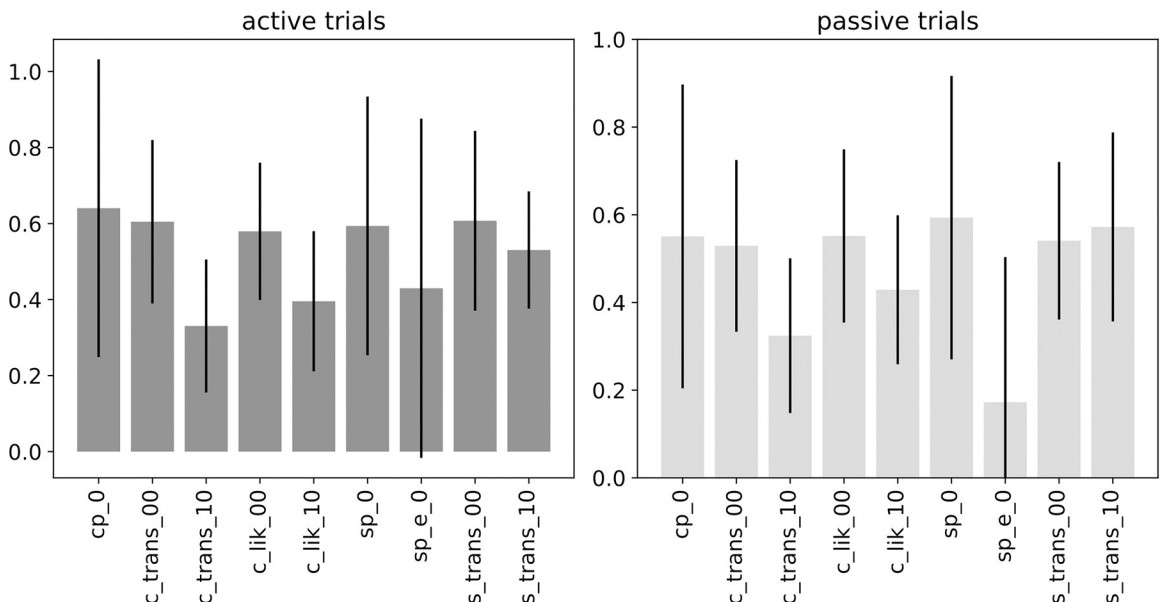

**Fig 10. Averaged parameters for common- and separate cause models (Experiment 2).** Average parameter values for the optimized parameters of Experiment 2, separate for active (left panel) and passive trials (right panel). Parameter names: cp_0: common cause chain, prior on first state 0; c_trans_00: transition from state 0 to 0; c_trans_10: transition from state 1 to 0, c_lik_00: likelihood of inferring state 0 after observing 0; c_lik_10: likelihood of inferring state 0 after observing 1, sp_0: separate cause chain, prior on state 0, s_trans_00: transition from 0 to 0, s_trans_10: transition from state 1 to 0. Error bars represent the standard deviation caused by variance in parameters between subjects.

(or $P(cause = separate) > P(cause = common)$) for each experimental delay. When fitting a psychometric function to the empirical and simulated data, we find a good match between the two functions (Fig 11).

A model comparison yielded an advantage for the full causal model over all competing models, namely, a bias model, a forced fusion- and a forced separation model. The superiority of the full causal model held in both active- and passive movement trials (Fig 12). The evidence

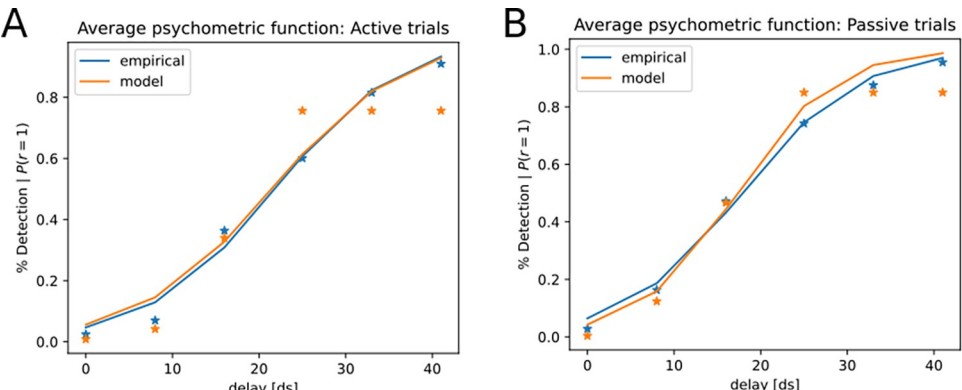

**Fig 11. Average empirical vs. model-predicted psychometric functions for active and passive trials (Experiment 2).** Psychometric functions were fitted to empirical detection probabilities per delay (blue) vs. detection probabilities predicted by the model (orange). **A.** Active trials **B.** Passive trials.

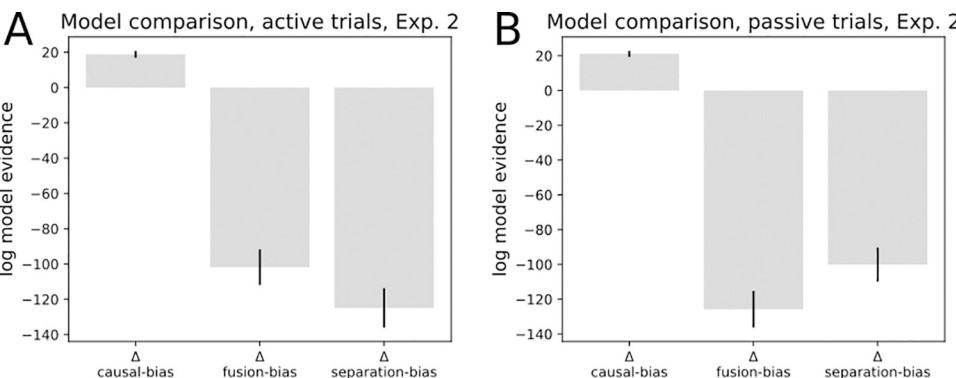

**Fig 12. Model comparison, Experiment 2. A.** Model comparison on active movement trials: a full causal model is compared to a bias-only model (causal-bias). Positive values indicate higher model evidence for the full causal model. The bias model is superior to a forced fusion model (fusion-bias), and a forced separation model (separation-bias). **B.** Model comparison on passive movement trials. The full causal model is superior to a bias-only model. The bias-only model is superior to a forced-fusion and a forced-separation model.

difference per participant ($\Delta$ LAP causal–LAP bias) in active movement trials was 18.82 ($\pm$ standard error 1.98), indicating an advantage for the full causal model over a bias-only model on the participant level. The advantage approached certainty on the group level ($1.62e^{196}$) The bias-only model held an advantage over the forced fusion model ($\Delta$ LAP fusion—LAP bias -101.80 $\pm$ 10.06). The bias model was superior to a forced separation model ($\Delta$ LAP separation —LAP bias: -124.87 $\pm$ 11.15). In passive movement trials, the causal model was superior to a bias model ($\Delta$ LAP causal–LAP bias: 21.07$\pm$1.73). The bias model was superior to a forced fusion model ($\Delta$ LAP fusion–LAP bias: -125.77$\pm$10.54) and to a forced separation model ($\Delta$ LAP separation–LAP bias: -100.08$\pm$9.76).

We finally performed separate optimizations for the active- and passive trials. This optimization yielded distinct participant-specific sets of parameters for the two conditions. When simulating the psychometric function with the active- vs. passive condition parameters, our model replicates the finding of an increased threshold in the active- compared to the passive condition (Fig 13), capturing empirical patterns of sensory attenuation in active trials in Experiment 2. This distinction is driven by subtle changes in the parameters of the generative model between active and passive trials (Fig 10). As a measure of global fit of the full model, the mean KL divergence for Experiment 2 was $KL_{mean}$ = 3.396 nats (Fig 14B). Two outlier participants are related to a lower-than-average detection performance (KL-divs = 10.484, and 13.267; see S3 in S1 File for details) The model's predictions could be successfully recovered by our fitting approach (see supplementary chapter S2, S16, S17 Figs in S1 File).

## Discussion

Sensory attenuation is a ubiquitous phenomenon in sensory processing across species. Theoretically, it has been explained with mechanisms of unspecific sensory gating [65] or predictive processes via efference copies generated from forward models [9]. Recently, sensory attenuation has been framed as a consequence of Active Inference [32, 42]. We here expand on the recent computational account of sensory attenuation by adding the perspective of Bayesian Causal Inference (BCI, [48]). Across two different experimental datasets in the sensorimotor (*Experiment 1*) and visuo-proprioceptive (*Experiment 2*) modalities, we show that our model based on BCI by sum-product message passing in a graphical network can capture patterns of sensory attenuation qualitatively in simulations, and yields a good quantitative fit to the data.

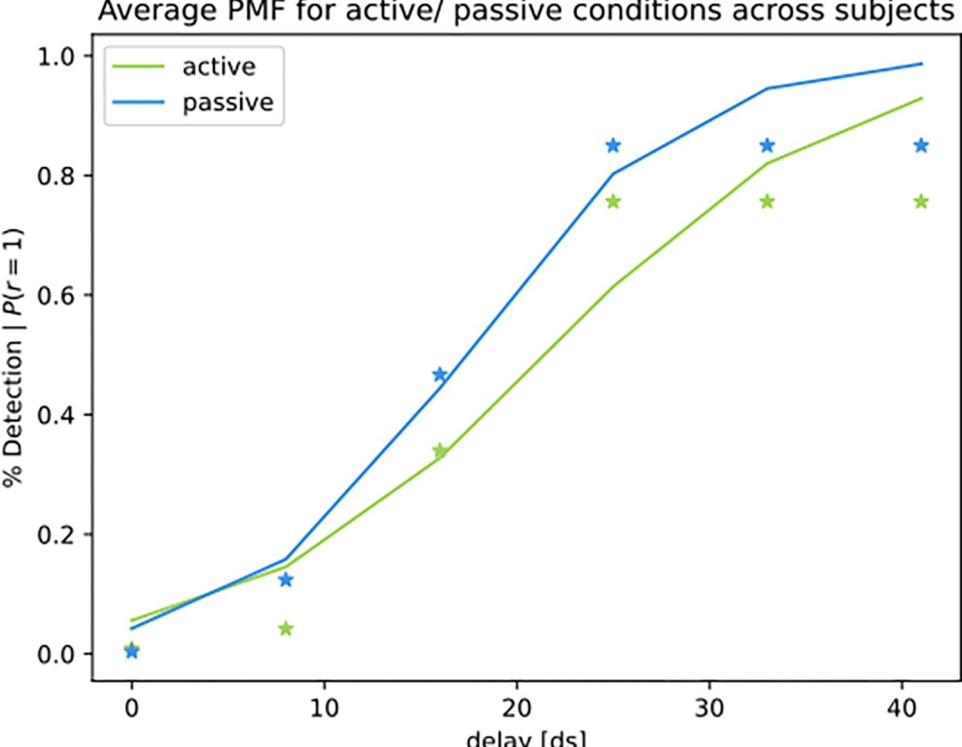

**Fig 13. Simulated average psychometric function for active vs. passive trials (Experiment 2).** Average psychometric function for active vs. passive trials, averaged across all participants.

*Experiment 1* investigated sensory attenuation based on sensorimotor predictions [23]. A key finding was that tactile sensitivity on the moving finger is more strongly reduced for tactile probes that were congruent than incongruent to the sensorimotor predictions. The BCI model can reproduce this key finding: an internal cause is inferred for prediction-congruent probes. The brain constantly tries to maximize the information about its surroundings. Since self-generated sensations are not caused by unexpected changes due to external, environmental factors, these sensations are processed in an attenuated manner [66]. When fitting the model to the data with optimized parameters for each participant, empirical patterns are particularly well captured across trials where probes were incongruent with predictions. The model's predictive power is slightly reduced for trials in which probe and object frequency were congruent. Note that the model's ability to capture these behavioral effects might be due to relatively small empirical differences between the experimental conditions. To best test our model in the context of tactile sensory attenuation, it may require clearer differences between experimental conditions on the behavioral level. Our model nevertheless contributes an improved understanding of prediction-congruent sensory attenuation in the tactile domain as the first direct application of BCI to individual, implicit measures of sensory attenuation.

*Experiment 2* was concerned with the detection of delays between visual and proprioceptive information during a guided arm movement [6]. During active movement, delay detection performance decreased compared to trials where the participants' arm moved passively. We first reframe the task as the problem of inferring common- or separate causes for the visual (video feedback) and proprioceptive information from the arm. With increasing delays, the participant should become more and more certain that separate causes are underlying the visual- vs. the proprioceptive information, while after observing small delays between the

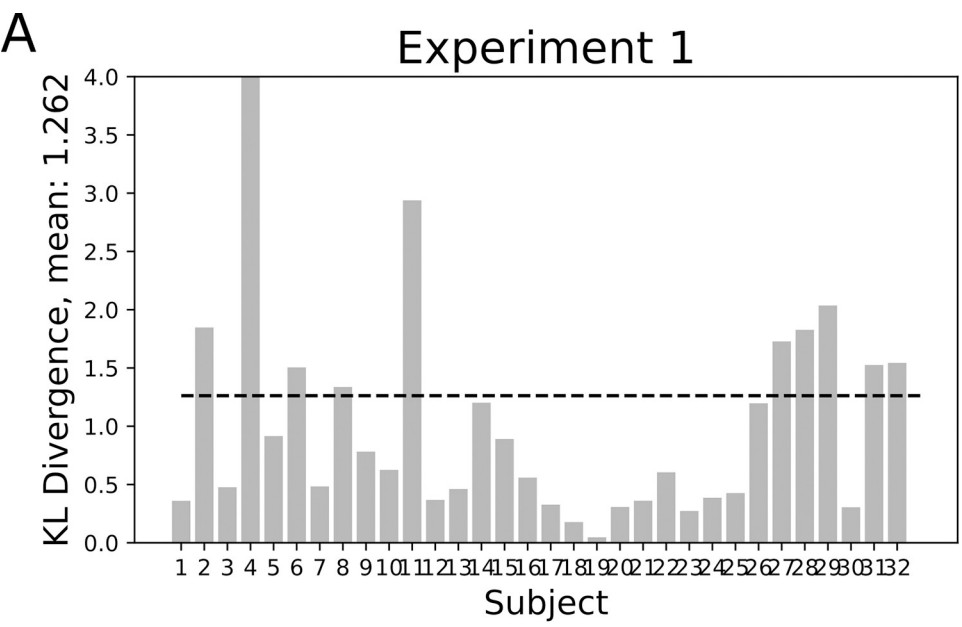

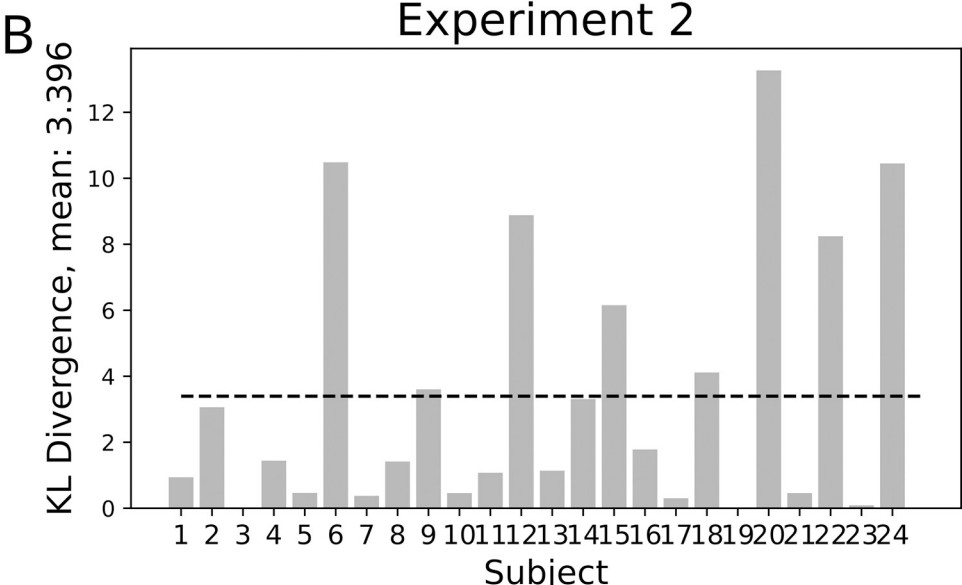

**Fig 14. Global fit (KL-divergence), subject-specific. A.** Subject-specific KL-divergence between model-derived and empirical response distributions, Experiment 1, mean divergence was $KL_{mean}$ = 1.262 nats **B.** Experiment 2, with mean $KL_{mean}$ = 3.396 nats.

visual and proprioceptive sensory streams, the participant should favor a common cause for the visual and proprioceptive input. We reproduce this general pattern by showing that a common cause model is preferred for trials with smaller experimental delays, whereas a separate cause model is preferred after observing simulated data with larger delays. Fitting the model to the data with optimized parameters yields a good fit between empirical and model-derived psychometric functions. We furthermore found that when fitting the model to trials with an active- vs. a passive movement and simulating task behavior separately for each trial type, our model can capture the empirical finding of increased sensory attenuation in active- compared to passive trials.

A strength of the present model lies in its direct, individual-level representation of the computational mechanisms underlying sensory attenuation. While we here show that the model can capture empirical signatures of sensory attenuation, it is flexible enough to model data from different experimental paradigms and perceptual modalities. Since sensory attenuation is assumed to be a crucial and domain-general function of sensory information processing, a model based on variational BCI is a suitable choice to investigate the computational mechanisms underlying sensory attenuation. Unlike descriptive approaches, computational models of central cognitive functions can deepen our understanding of the multisensory processes involved in the phenomenon of sensory attenuation. Classical theoretical frameworks for sensory attenuation come with some crucial limitations that cannot account for the attenuation of externally generated stimuli which are likely not part of a forward model [32] or contradictory results that run against the predictions of simple forward model accounts [67]. For example, van Kemenade and colleagues (2016) have provided evidence for multisensory facilitation in a delay detection task, i.e., participants showed improved performance in trials with bimodal action consequences vs. unimodal. In our model, bimodal trials correspond to two streams of sensory information connected to the model's hidden state; improving detection performance by means of the availability of more sensory information about the hidden state available for inference. In the future, our BCI model of sensory attenuation may be applied to gain a more mechanistic understanding of multisensory facilitation.

Our BCI model for *Experiment 2* is closely related to Bayesian models of body ownership [53, 55], where e.g. the rubber hand illusion is strengthened when a common cause for visual, tactile and proprioceptive information can be inferred [55]. Sensory attenuation appears to play an important role for body ownership and agency, as supported by aberrant patterns of sensory attenuation in individuals with psychosis [34, 68, 69] and individuals with increased levels of psychosis proneness [36]. The deficit in predicting the sensory consequences of one's own actions in psychosis may stem from disturbances in the contextualized regulation of sensory precision, mediated by dopamine imbalances [32]. Similarly, patients with psychogenic functional movement disorder show aberrant sensory attenuation [70]. These findings suggest that sensory attenuation is not a quirk, but a crucial aspect of motor control. Understanding the computational and neurobiological underpinnings of sensory attenuation may be a fruitful avenue towards novel therapeutic approaches to disturbances of body ownership and motor control.

## Future directions

The present work provides a starting point for computational descriptions of sensory attenuation. Our model's sequential nature offers potentially high temporal resolution of intra-trial dynamics, which are here inferred based on dichotomous behavioral responses. At this point, we would not claim that the model's parameter values reflect neuronal variables. Rather, we offer an interpretation of sensory attenuation as a causal inference process. However, sensory attenuation shows interesting dynamical patterns in motion capture [28] and EEG [71] which we can account for in a sequentially structured model. The BCI model's fit with these phenomena needs to be investigated in future studies. While the model passed initial validation procedures, future effort needs to be directed towards thorough testing and empirical validation of our model. For instance, experimental work is necessary to study how precisely model parameters are affected by specific experimental manipulations. The test-retest reliability of individual-level parameters across time and across experiments needs to be established. If successful, the BCI model could be applied to study altered sensory attenuation in clinical populations. As previously mentioned, failures of sensory attenuation have been connected to false inferences

about agency and body ownership in schizophrenia [32, 34–37]. In future efforts, our computational model may be expanded towards formalizing aberrant Causal Inference in patients with psychosis or heightened levels of psychosis proneness. When applied to data from clinical populations, variational BCI may help deepening our understanding of the computational mechanisms underlying decreased sensory attenuation in individuals with psychosis. If replicated, the neurobiological basis of sensory attenuation deficits in psychosis can be guided by computational markers derived from our model. Secondary somatosensory cortex and the right temporoparietal junction (TPJ) can serve as a candidate ROI [68, 72]. However, reliable neural- and computational-level profiles distinguishing healthy controls from individuals with schizophrenia could be more challenging than it appears from the current literature (i.e., publication bias). In a recent study using flash-beep stimuli, medicated patients with schizophrenia showed neurocomputational signatures of Bayesian causal inference mostly comparable to healthy controls [73]. For further validation of our model, and for an extension towards clinical populations, it is crucial to identify reliable experimental paradigms that produce robust inter-individual differences in causal inference- and sensory attenuation. Voss and colleagues [37] have further demonstrated that sensory attenuation can be manipulated by applying repetitive TMS pulses over primary motor cortex. If future studies can uncover a neurobiological basis for the parameters governing variational BCI, their manipulation using TMS could be of high scientific and therapeutic interest. Using transcranial direct current stimulation (tDCS) improvements in delay detection [74] and delay adaptation [75] had been already shown in patients with schizophrenia spectrum disorders. However, the BCI could improve our understanding of which parameter exactly contributed to these initial findings.

Similarly, aberrant sensory attenuation has been reported for individuals with functional psychogenic movement disorders [70]. Understanding the computational and neurobiological basis governing aberrant movement patterns in this population may generate novel hypotheses and be of interest for clinical applications in the future.

## Conclusion

A range of cognitive functions has been successfully described by BCI, including multisensory perception, motor learning and body ownership. We here presented a model based on variational message passing and BCI that captures empirical patterns of sensory attenuation across different experimental paradigms and perceptual modalities. The model provides individual-level, quantitative predictions on psychophysical aspects of sensory attenuation and achieves a good quantitative fit with group-level behavioral markers. We accommodate a core function of motor behavior within a unifying, neurobiologically plausible computational framework, where sensory attenuation stems from actively inferring an *internal* cause for sensory information.

## Supporting information

**S1 File. Supporting information.** Supplementary materials and analyses on belief propagation, parameter recovery, outliers, effect of T.
(DOCX)

## Author Contributions

**Conceptualization:** Anna-Lena Eckert, Elena Fuehrer, Christina Schmitter, Benjamin Straube, Katja Fiehler, Dominik Endres.

**Data curation:** Elena Fuehrer.

**Formal analysis:** Anna-Lena Eckert.

**Funding acquisition:** Benjamin Straube, Katja Fiehler, Dominik Endres.

**Investigation:** Elena Fuehrer, Christina Schmitter, Benjamin Straube.

**Methodology:** Anna-Lena Eckert, Elena Fuehrer, Christina Schmitter, Benjamin Straube, Katja Fiehler, Dominik Endres.

**Project administration:** Anna-Lena Eckert.

**Resources:** Benjamin Straube, Katja Fiehler, Dominik Endres.

**Software:** Anna-Lena Eckert, Dominik Endres.

**Supervision:** Benjamin Straube, Katja Fiehler, Dominik Endres.

**Validation:** Dominik Endres.

**Visualization:** Anna-Lena Eckert.

**Writing – original draft:** Anna-Lena Eckert.

**Writing – review & editing:** Anna-Lena Eckert, Elena Fuehrer, Christina Schmitter, Benjamin Straube, Katja Fiehler, Dominik Endres.

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
