## [Decision Letter · Decision Letter 0]

24 May 2024

PONE-D-24-14292Modelling sensory attenuation as Bayesian causal inference across two datasetsPLOS ONE

Dear Dr. Eckert,

Thank you for submitting your manuscript to PLOS ONE. After careful consideration, we feel that it has merit but does not fully meet PLOS ONE’s publication criteria as it currently stands. Therefore, we invite you to submit a revised version of the manuscript that addresses the points raised during the review process.

 Your manuscript has been reviewed by two experts in the type of modeling approaches that you use in your study, and you will find their detailed comments below. Both reviewers identified two related main issues which would need to be addressed and which might require additional analyses. First, I agree with the reviewers that the number of free parameters in your models (12 in Exp. 1 and 20 in Exp. 2) is rather large and might have resulted in overfitting. As pointed out by Reviewer #2, this problem might be exacerbated by the lack of a strong comparison model. Second, a parameter recovery should be performed and reported in the manuscript. Should you choose to submit a revision, please also make sure that the analysis/modeling code is made available (access to the Gitlab link that you provided appears to be restricted to members of your institution).

We look forward to receiving your revised manuscript.

Kind regards,

Patrick Bruns

Academic Editor

PLOS ONE

“This work was supported by the cluster project “The Adaptive Mind”, funded by the Excellence Program of the Hessian Ministry for Science and the Arts.”

3. We noted in your submission details that a portion of your manuscript may have been presented or published elsewhere. [Fuehrer et al., 2022, PNAS; Uhlmann et al., 2020, Hum Brain Mapp] Please clarify whether this [conference proceeding or publication] was peer-reviewed and formally published. If this work was previously peer-reviewed and published, in the cover letter please provide the reason that this work does not constitute dual publication and should be included in the current manuscript.

Reviewers' comments:

Reviewer's Responses to Questions

**Comments to the Author**

1. Is the manuscript technically sound, and do the data support the conclusions?

Reviewer #1: Yes

Reviewer #2: Yes

2. Has the statistical analysis been performed appropriately and rigorously? 

Reviewer #1: Yes

Reviewer #2: Yes

3. Have the authors made all data underlying the findings in their manuscript fully available?

Reviewer #1: No

Reviewer #2: No

4. Is the manuscript presented in an intelligible fashion and written in standard English?

Reviewer #1: Yes

Reviewer #2: Yes

5. Review Comments to the Author

Reviewer #1: In order to explain sensory attenuation phenomenae, the authors of this stude develop a model based on the Bayesian Causal Inference model, that infers whether a signal was self generated or external. Specifically they model two previously published experiments, showing that the model can at least qualitatively explain some of the observed effects.

This is a technically challenging model, and I appreciate the efforts that have gone into this.

The paper is well written, although given the complexity of the model I do worry how much can be learned from it (see more below).

Models for both experiments have a lot of free parameters to fit (even 12 is a lot). I would be very worried about the model being over specified, hence over fitted. I would be very curious to see a parameter recovery excercise. I.e. if simulated data were drawn from a model with a specific set of parameters, how well could these parameters be recovered when doing the model fitting as in the manuscript (but now to simulated data for which you know the true parameters). Even if the fitted parameters are not of interest, there would always be a worry that the model is not addressing a real aspect of the data, and that any sufficiently detailed model (with e.g. 20 parameters) would have been able to capture the data. I realise that doing more simulations would be time consuming, but I think the authors should find a way to address this concern.

How sensitive is the models to parameter choices? E.g number/length of time steps

What is the actual transition distribution described in the methods for Exp 1? I could not find it. Is it just a 2x2 transition matrix? If so it should be mentioned in Methods

Please insert the figures at the appropriate places in the manuscript. No journal is draconian enough to still want figures at the end.

L405 remove duplicate ‘for the’

Reviewer #2: Eckert and colleagures present a computational study to offer a novel explanation for the well-known phenomenon of sensory attenuation: The study develops a Bayesian causal inference (BCI) model to explain sensory attenuation of predictable sensory inputs because they were internally self-generated by motion (i.e. arise from a common signal cause as prioprioceptive signals) in contrast to externally attributed sensory inputs that are not attenuated because they inferred to come from an independent cause. The study develops the BCI model for two motion tasks and validates the BCI model on two previously published data sets using simulations and individual fits to the data. The authors report a decent fit of the model to the data and the model approximately fits important effects in the behavioral data.

While the theoretical contribution of the work is novel and important (i.e., explain sensory attenuation by BCI), I have many concerns about the computational modelling and presentation of results. Specifically, many details on the model formulation and fitting are lacking so that I doubt it would be possible to independently replicate the analyses (which I find especially important when introducing a novel model; see my comments below). Additionally, given the relatively few experimental conditions, the models appear highly complex with many free parameters whose meaning is unclear (i.e., difficult to interpret as indices of some cognitive mechanisms). Further, the manuscript needs some quality checks to increase the readability, especially on the figure and figure legends.

My more specific comments:

Intro

• The intro provides three important accounts for sensory attenuation, i.e. efference copies, active inference and Bayesian causal inference. It may be a matter of style, but the links between the three accounts appears quite loose (e.g. which and how can shortcomings of efference copy account can be explained by active inference / BCI?).

Methods

• Exp. 1 manipulation check: Did participants stroke with the targeted motion speed of 203mm/s that influences the intended frequency of stimulation? This might have been presented in the original publication, but should be referenced here.

• In the Methods description of Exp. 2 and Fig. 1B the experimental factors are not consistently reported: The overall design manipulated 2 agency X 2 hand identity X video delay (0 – 417 ms)?

• Justification and details of the BCI model:

o Exp.1: It is unclear why the task is modelled with sequential BCI using an HMM: Even though the stimulus is extended in time, a “classical” BCI would assumed “accumulated / final” percepts for the frequency of haptic self-stimulation and the frequency of external tactile stimulation and compute the two potential causal structures from these likelihoods (and the prior; cf. Koerding 2007). Thus, it is unclear why the dynamics of stimulation are modelled here, especially when the behavioral response depends on a final “static” perceptual estimate. While modelling the dynamic might be interesting for future studies (e.g. EEG), this seems to add a lot of complexity here. This questions also holds for exp. 2.

o Fig. 3B and text: It is unclear how “experimental manipulations were translated into a series of frequency observations”. From which model variables (e.g., likelihood distributions, probability estimates for external event) were the frequency observations derived? Fig. 3B axis labels suggest that frequency is changing as a function of time which was clearly not manipulated in the experiment…

o The model parameters (e.g. mean and variance of likelihood distribution) and their meaning are not clearly described (e.g., in an overview). In exp. 2 there are seemingly 20 (!) parameters, and it is unclear where they come from exactly and what they mean. This is later partially mentioned in the results, but I think the reader should already get an overview of parameters in the methods.

o H0 / H1 could be explained in more detail. H0 is a very weak competing model, i.e. uniform random choice of the three possible hidden states. Are there more powerful alternative models beyond causal inference (e.g., classically, BCI models are compared to forced-fusion and forced-segregation models)?

o Overall, details are missing how the generative model in exp. 1 and 2 is linked to inferred internal states/estimates and is finally linked to behavioral responses (e.g. how is p(ext) finally estimated? What is exactly the likelihood function in exp. 2?)

o Was model fitting in Exp. 1 repeated with multiple parameter initializations to check stability of the results (in exp. 2 at least 2 initializations)?

o It is actually good practice in computational modelling (e.g. Wilson & Collins, 2019) to perform parameter recovery (and even model recovery) to show that the parameter are reliably estimated with the used fitting procedure. Especially for a novel computational model, this appears to be important.

o In exp. 2, the model is separately fitted to active vs. passive movement, I assume by refitting all 20 model parameters. This effectively doubles the number of parameters to fit a single experimental observations, which is not a very sparse way to explain the effects. Is the distinction active vs. passive not better captured by a single parameter with a meaning (e.g. a sensory variance)?

• Fig. 3C: x/y axis legends missing.

Results

• Fig. 4: Why does the attribution of internal/external cause appears only transiently, but after “object contact” always shifts to internal? From figure 6 I can learn that around this time the probe is presented, but how does this affect the model when the haptic signal is actually only presented later (after object contact)? This needs better explanation. (Please also check legends A, B, C – assignment wrong).

• Fig. 5A: The model’s replication of the behavioral congruency effects (which also appear rather weak) is not very good – how good is the overall model fit (e.g. explained variance / coefficient of determination)?

• The results present first simulations, then partially individual results with individual parameter fits. It is unclear why not simply average results at group level are presented with individually fit parameters – that would increase the readability of the manuscript.

• Bernoulli factors in exp. 1 and 2: Bernoulli factor compare the fitted model to a non-fitted model with fixed parameters, but do Bernoulli factors take model complexity into account? E.g. Bayesian information criterion penalizes free parameters when comparing the fit of two models…and the null model has 0 and the main model has 12 / 20 (?) free parameters. I.e. the null model needs a fairer comparison and the author might consider a stronger comparison model (i.e. forced fusion /segregration)

• I have a hard time to match Figure 8A and its legend…what is the figure showing exactly? PSE and relative model evidences?

• Model reproduction of the threshold-shift for active vs. passive movement: As stated in the comment above, it appears highly unsparse to refit a full set of parameters to explain a single experimental manipulation. As a result, the effect of the manipulation cannot be attributed meaningfully to parameters (Supplemental fig. 16). The authors might consider which of the model’s parameters is likely affected by an active/passive manipulation, and more systematically analyse which parameter can explain the effect (e.g. priors vs. sensory variances).

• Fig. 12: Again, it should be made clearer that A is participant data and B is model prediction. It is unclear why 6 participants are removed from B, appears somewhat as an adhoc choice.

Discussion:

• I agree that the developed BCI models would be an attractive option to quantify psychopathological abberations e.g. in schizophrenia, but I would add the caveat that the model first needs more basic testing and experiments before such an effort is fruitful (e.g. parameter recovery as a start, test-retest reliability of parameters, specific experiments to explore which experimental manipulations affect which parameters etc). The models are an important starting point, but they are also very complex (i.e., many parameters and estimation of dynamics) and need more validation in future studies.

6. PLOS authors have the option to publish the peer review history of their article (what does this mean?). If published, this will include your full peer review and any attached files.

Reviewer #1: No

Reviewer #2: No

---

## [Author Response · Author response to Decision Letter 0]

23 Aug 2024

Dear reviewers, 

a point by point response to all your thoughtful and constructive comments can be found in the attached file "Response to reviewers". 

Thank you and kind regards

---

## [Decision Letter · Decision Letter 1]

24 Sep 2024

PONE-D-24-14292R1

Modelling sensory attenuation as Bayesian causal inference across two datasets

PLOS ONE

Dear Dr. Eckert,

Thank you for submitting your manuscript to PLOS ONE. After careful consideration, we feel that it has merit but does not fully meet PLOS ONE’s publication criteria as it currently stands. Therefore, we invite you to submit a revised version of the manuscript that addresses the points raised during the review process.

As you will see below, while both reviewers appreciated the revisions to the manuscript, they were also concerned about the result of the parameter recovery, in particular about the substantial variability as well as some rather large differences between empirical and recovered parameters. Both reviewers recommend to visualize the correlation between empirical and recovered parameters in a scatter plot instead of only showing the distributions. Based on the results that are shown in Figs. S14 and S15, however, there are major concerns about the reliability of the model fittings which need to be addressed. Reviewer #1 makes very clear recommendations in this regard.

We look forward to receiving your revised manuscript.

Kind regards,

Patrick Bruns

Academic Editor

PLOS ONE

Reviewers' comments:

Reviewer's Responses to Questions

**Comments to the Author**

1. If the authors have adequately addressed your comments raised in a previous round of review and you feel that this manuscript is now acceptable for publication, you may indicate that here to bypass the “Comments to the Author” section, enter your conflict of interest statement in the “Confidential to Editor” section, and submit your "Accept" recommendation.

Reviewer #1: (No Response)

Reviewer #2: All comments have been addressed

2. Is the manuscript technically sound, and do the data support the conclusions?

Reviewer #1: No

Reviewer #2: Yes

3. Has the statistical analysis been performed appropriately and rigorously? 

Reviewer #1: No

Reviewer #2: Yes

4. Have the authors made all data underlying the findings in their manuscript fully available?

Reviewer #1: Yes

Reviewer #2: Yes

5. Is the manuscript presented in an intelligible fashion and written in standard English?

Reviewer #1: Yes

Reviewer #2: Yes

6. Review Comments to the Author

Reviewer #1: The updated version of the manuscript includes a lot moire details on the modeling, a parameter recovery excercise and some more clarifications to the introduction. I very much like the inclusion of the graphical model in Fig 2

I appreciate all the work the authors have done, but unfortunately I am now even more worried about the reliability of the model fittings.

First (and least worrying), when doing parameter recovery it is better practice to sample (uniformly or randomly) from a set of model parameter, as opposed to just choosing the ones that have already been optimised. The reason being that if there is problem with the optimization then the process might just recreated the same values again. As a toy example you could imagine an parameter fitting process that always returns alpha_empirical=3.14159. Using that value as starting point of a parameter recovery (response simulations, parameter recovery), one should not be surprised to recover alpha_simulated=3.14159. It is unlikely to be a major issue with multiple subjects fittings though.

Hence that is not the major problem here.

Another issue is that in Figure S14 and S15 the distribution of parameters are compared, as opposed to the individual subject parameters. This could be better illustrated with a scatter plot, with points hopefully close to the identity line. The problem with just plotting the two distribution is of course that it does not show the correlation of the values, or the difference between ‘empirical’ and ‘simulated’.

However the biggest problem is probably that Fig S14 and S15 looking at the distributions it is seems that there are very big differences between the ‘empirical’ and ‘simulated’. E.g. cp_0 in Fig. S15. ‘Outliers’ in recovered parameters may be part of the problem, but it is hard to tell from the plot of distributions (see point above).

I am glad to see that the number of parameters were reduced but I am now worried that the parameter recovery has essentially failed, i.e. that the model fitting is not working, due to overfitting. In that case any fitted parameters values are not to be trusted, and the model should just be described as a way to fit the data out of many but that it is not likely to actually reflect any processes in how the brain is actually performing the task (as an example you can also fit the data with a large enough neural network without any time dynamics, but we would not think that is how the brain solves the problem).

Based on this problem I see two ways out: The authors acknowledging the problems I have mentioned above (which would be a major limitation on the value of the paper), or the authors find a way to restrict the model further so that the parameter recovery procedure works (which could be a big piece of work).

(Suggestion that may not be useful: it may be worthwhile to look at scatter plots of fitted parameters, If any pairs of parameters are highly correlated it implies that one of them is redundant)

Regardless, I think the authors should change the plot of the parameter recovery in S14 and S15 to show the correlation or differences in fitting parameters (as above).

In Fig S14 some of the error bars are horizontal? Also, I thought some of these parameters had been removed but still appear in the plot? (e.g. p_none)

In Fig 3 what do the numbers in the upper part of the figure refer to? If parameters, should there not just be 9 (including sigma) in A, 8 in B?

Likewise Fig 6 shows 11 parameters? Have these not been updated? Or are they just shown for visualisation purposes (in which case p_none eg is not actually a fitted parameter)

Fig 6 C: Mention that this is a violin plot and put sigma on the vertical axis

Suggestion: In Fig 7 it could be useful to have a small horizontal bar at the bottom to indicate the probe period

Line 414, Error!

Reviewer #2: Overall, the authors addressed my concerns very well and substantially improved the manuscript.

I have only two small remaining concerns, but specifically the author now:

• improved the coherence of the Introduction.

• more accurately describe the procedures (participant stroking) and experimental design.

• motivate the usage of an HMM to implement a sequential BCI model, better explain the model parameters and how the model estimates are linked to behavioral responses, visualize the model and add modeling details for better understandability/replicability.

• changed the model architecture to reduce the number of parameters considerably (esp. for exp. 2), which even appears to improve the model fit and predictions. Further, they introduced stronger competing model (bias model, forced fusion, forced segreation) which are still outperformed by the original BCI model.

• implement and report results of parameter recovery in Fig. S5 / S6. Yet, they only compare original and recovered parameters on average. It would be more informative to plot original vs. recovered parameters for each individual (scatter plot) so that not only average bias can be accessed but also individual consistency (e.g. quantified by a correlation or ICC). Quite large error bars on the recovered parameters suggest that individual recovered parameters could have substantial variability (even if accurate on averge).

• still do not report a measure of overall model fit – would that be possible?

• rather report group level data and model fits than individually selected participants.

• may consider a very recent publication (https://journals.plos.org/plosbiology/article?id=10.1371/journal.pbio.3002790 ) in their discussion how their computational model can capture aberrant causal inference in schizophrenia.

7. PLOS authors have the option to publish the peer review history of their article (what does this mean?). If published, this will include your full peer review and any attached files.

Reviewer #1: No

Reviewer #2: No

---

## [Author Response · Author response to Decision Letter 1]

29 Nov 2024

Dear reviewers, 

thank you for taking the time to provide constructive and detailed feedback throughout the review process. Please find our updated manuscript and a point-by-point response letter attached. 

Thank you and best wishes

---

## [Decision Letter · Decision Letter 2]

8 Jan 2025

Modelling sensory attenuation as Bayesian causal inference across two datasets

PONE-D-24-14292R2

Dear Dr. Eckert,

We’re pleased to inform you that your manuscript has been judged scientifically suitable for publication and will be formally accepted for publication once it meets all outstanding technical requirements.

Kind regards,

Patrick Bruns

Academic Editor

PLOS ONE

Additional Editor Comments (optional):

Reviewers' comments:

Reviewer's Responses to Questions

**Comments to the Author**

1. If the authors have adequately addressed your comments raised in a previous round of review and you feel that this manuscript is now acceptable for publication, you may indicate that here to bypass the “Comments to the Author” section, enter your conflict of interest statement in the “Confidential to Editor” section, and submit your "Accept" recommendation.

Reviewer #1: All comments have been addressed

2. Is the manuscript technically sound, and do the data support the conclusions?

Reviewer #1: Yes

3. Has the statistical analysis been performed appropriately and rigorously? 

Reviewer #1: Yes

4. Have the authors made all data underlying the findings in their manuscript fully available?

Reviewer #1: Yes

5. Is the manuscript presented in an intelligible fashion and written in standard English?

Reviewer #1: Yes

6. Review Comments to the Author

Reviewer #1: Thank you for the changes made. It took me a little while to understand the Prediction recovery now used (BTW maybe also change section S2 title?) but it makes sense now.

My only other comment is that figure S15 now needs to be rotated (or A B C D placed on columns)

7. PLOS authors have the option to publish the peer review history of their article (what does this mean?). If published, this will include your full peer review and any attached files.

Reviewer #1: No

---

## [Editor Report · Acceptance letter]

13 Jan 2025

PONE-D-24-14292R2 

PLOS ONE

Dear Dr. Eckert, 

I'm pleased to inform you that your manuscript has been deemed suitable for publication in PLOS ONE. Congratulations! Your manuscript is now being handed over to our production team.

Kind regards, 

on behalf of

Dr. Patrick Bruns 

Academic Editor

PLOS ONE